# Optimization and Quality Assessment of Arrival Time Picking for Downhole Microseismic Events

**DOI:** 10.3390/s22114065

**Published:** 2022-05-27

**Authors:** Jiaxuan Leng, Zhichao Yu, Zhonghua Mao, Chuan He

**Affiliations:** 1School of Earth and Space Sciences, Peking University, Beijing 100871, China; jxleng@pku.edu.cn; 2National Supercomputing Center (Shenzhen Cloud Computing Center), Shenzhen 518055, China; zcyu.426@163.com; 3Shengli Branch, Geophysical Company, SINOPEC, Dongying 257086, China; wtmzh@126.com

**Keywords:** downhole microseismic monitoring, waveform similarity coefficient, iterative cross-correlation, arrival optimization, quality assessment

## Abstract

Arrival-time picking is a critical step in microseismic data processing, and thus the quality control of arrival results is necessary. Conventional picking methods may be inaccurate or inconsistent due to varied signal-to-noise ratios (SNR) and waveform patterns of the events recorded in different time sections. To address this issue, we propose a quality assessment method based on waveform similarity coefficients to evaluate arrival results and also a global optimization algorithm based on iterative cross-correlation to refine arrival times. The recordings after moveout correction are applied to calculate the intra-event and inter-event waveform coefficients for the quality assessment of arrival results. The residual time differences of intra-event and inter-event traces are calculated sequentially using an enhanced iterative cross-correlation method. In addition, the stacked waveform of each event after the intra-event residual time correction is introduced for global optimization to obtain the inter-event residual time discrepancies. We use both synthetic data and field data to validate the proposed method. The results indicate that the proposed method yields more robust and reliable results. The quality assessment of the optimized arrivals is greatly enhanced compared to the adjusted picks obtained from single event-based processing methods.

## 1. Introduction

Microseismic monitoring technology captures and analyzes detectable elastic waves generated by rock ruptures and has been widely used in the hydraulic fracture stimulation of unconventional reservoirs [1,2,3] and rock/mining engineering [4,5]. Arrival-time picking is a crucial step in microseismic data processing, and the picking result is useful for phase identification, hypocenter localization, mechanism analysis, and facture interpretation [6]. Some benchmarking studies have revealed that significantly different results in microseismic interpretations can occur for the same data set [7]. Therefore, it is essential to include quality control (QC) procedures during data processing and analysis to ensure more internally consistent results, which will ultimately help to optimize stimulation programs and production [8,9].

In recent decades, various algorithms have been substantially developed in earthquake and exploration seismology to detect and pick the arrivals of different seismic waves from 1-C or 3-C recordings. Akram and Eaton reviewed the existing arrival-time picking algorithms on microseismic data processing, classifying them as single-level, hybrid, and multilevel-based methods [10]. A characteristic function is generally constructed from the single receiver recording in single-level algorithms, and the maximum of the characteristic difference is chosen as the arrival time of the microseismic signal. The short- and long-time average ratio method (STA/LTA) [11,12], Akaike information criterion method (AIC) [13], polarization-based method [14], higher-order statistics such as skewness and the kurtosis-based method [15,16], and the time-frequency analysis method [17] are commonly used single-level methods. Hybrid algorithms have also been proposed to achieve more accurate and precise arrival-time results for low signal-to-noise ratio (S/N) events by combining information from one or more individual picking algorithms [18,19]. With the rapid development of theoretical research on deep learning, some data-driven algorithms have also been proposed to determine the arrival times of microseismic events [20,21,22]. Multi-level algorithms make simultaneous use of information on multiple receiver levels within the array, and this kind of algorithm takes advantage of the similarity of the detected microseismic signals for different receivers and thus can improve the quality of picks. Estimating time delays among received signals is fundamental for multi-level algorithms. Several techniques are used to estimate the time delay, such as the cross-correlation method [23,24,25,26] and phase-only correlation method [27].

With the ever-increasing size of microseismic data volumes in real-time monitoring, a set of parameters is generally carefully selected to process all detected events in the arrival-time picking workflow. However, these automatic methods usually encounter a problem in a real-time-varying noisy environment. Arrival picks among events are inconsistent when the same judgment criteria are applied due to the differences in the SNR levels and waveform patterns. This type of inconsistency is usually ignored in conventional picking workflows and picking errors among events can hardly be revealed by analyzing individual event recordings. However, they can be observed as local inconsistencies if assessing the dataset as a whole and comparing recordings from different events. Therefore, the validation of the entire dataset and global optimization is necessary. The quality of P-wave and S-wave arrival is evaluated using microseismic multiplets [28]. Common-receiver gathers were implemented to identify wrongly picked arrival times and to analyze the quality of time picks [29]. Combining the receiver-oriented and the event-oriented approach is proposed to optimize joint arrival time picks for microseismic events [30].

In this study, we consider all microseismic event datasets as a whole and present a quality assessment method for evaluating arrival results as well as a global optimization algorithm for improving arrival picks on microseismic data. First, we introduce the quality assessment of the arrivals. Next, an iterative cross-correlation method with a probability density function is presented to calculate the residual time corrections of inter-receiver recordings. Then, global optimization is proposed to obtain the inter-event residual time discrepancies. Finally, we illustrate the performance of the proposed method in synthetic data and field data, demonstrating the effectiveness and superiority of our proposed method in consistency processing compared to the original picks and the adjusted picks obtained from single event-based processing.

## 2. Methodology

### 2.1. Quality Assessment of Arrival Time Picks

Hydraulic fracturing-induced microseismic events are often monitored by a downhole 3C sensor array deployed in one or more wells close to the treatment well. Microseismic events with nearby source locations and nearly identical focal mechanisms will show similar recordings [28,31]. Meanwhile, microseismic sources with the same focal mechanism, whether produced in the same fracturing stage or not, are expected to show similar traces recorded but distinct amplitudes at the same receiver. A microseismic multiplet is a group of seismic events with very similar waveforms but with different origin times. For the multiplets, the waveform similarity depends primarily on inter-event and inter-receiver distances and the time delay information is the key embodiment of inter-event distance, so the accuracy and consistency of arrival-time data are the targets of arrival picking.

Let *t_ik_* be the preliminary arrival time pick (P- or S-wave) for the *k*th event, recorded at *i*th receiver, and *t_jl_* corresponds to the preliminary arrival time pick for the *l*th event, recorded at the *j*th receiver. It is determined as the time lag for which the normalized cross-correlation function *cc_ikjl_* between the two seismogram traces *u_ik_* and *u_jl_* is maximum:(1)$$cc_{ikjl}(t)=\frac{\int u_{ik}(\tau )u_{jl}(\tau -t)d\tau }{\sqrt{\int u_{ik}^{2}(\tau )d\tau \times \int u_{jl}^{2}(\tau )d\tau }}$$

The correlation coefficient is the maximum value of *cc_ikjl_*, and Δ*t_ikjl_* is the time delay between two recordings. If the arrival time picks and the cross-correlation calculated relative arrival-times are identical, then they satisfy the following formula,
(2)$$t_{ik}-t_{jl}=\Delta t_{ikjl}$$

The measurements may not always be consistent due to varied noise interference or anomalous conditions of the monitoring equipment; thus the Equation (2) is not completely valid. When we construct the microseismic waveforms after moveout correction by arranging the arrival picking times into gathers to analyze the recordings, arrival errors can be observed as local discrepancies. The waveform similarity of the microseismic signals can be utilized to quantify a more precise residual time difference, which improves the accuracy and consistency of the arrival result. Supposing one record segment has the P- or S-wave arrival of a microseismic event in it, this phase should be aligned after an accurate moveout correction. The semblance coefficient is a very good measure of the arrival optimization because it shows a higher value if the waveforms of all the traces are aligned.

The similarity coefficient of inter-event multi-trace recordings is defined as
(3)$$S_{k}=\frac{\sum_{n=-N_{1}}^{N_{2}}{\left(\sum_{i=1}^{M}u_{ik}(t_{ik}+n)\right)}^{2}}{M\sum_{n=-N_{1}}^{N_{2}}\sum_{i=1}^{M}u_{ik}^{2}(t_{ik}+n)}.$$
where *u_ik_* is the recording of the *k*th event, recorded at the *i*th receiver, and *t_ik_* is the arrival time. *M* is the trace number. *N*_1_ and *N*_2_ are the time window length before and after the arrivals, respectively.

The stacking waveforms superposed by the recordings after moveout correction can be utilized as the representative signal for the microseismic events, and the similarity coefficient between them is a measurement for the consistency of the event arrivals. The similarity coefficient of two recordings is defined as
(4)$$X_{kl}=\frac{\sum_{n=1}^{N}{\left(w_{k}(n)+w_{l}(n)\right)}^{2}}{2\sum_{n=1}^{N}\left(w_{k}^{2}(n)+w_{l}^{2}(n)\right)}.$$
where *w_k_* and *w_l_* are the stacking waveforms after moveout correction of *k*th event and *l*th events, respectively, and *N* is the length of the waveform. The value of *S* and *X* lies between zero and one, and it is equal to one only if the waveforms of all the traces are completely the same in shape and amplitude [32]. In practice, the value will reach its maximum.

Figure 1 depicts two synthetic downhole microseismic events with different Gaussian white noise values and their recordings after move correction using arrivals with various error levels. We evaluate the arrivals by calculating the similarity coefficients (Table 1); *S*_1_ and *S*_2_ are the similarity coefficients of multi-trace recordings after the moveout correction of two events, and *X*_12_ is the similarity coefficient of two stacking waveforms. As the arrival error level increases, the waveform becomes inconsistent, and the similarity coefficients of the recordings after moveout correction decrease significantly. In other words, the reduction in residual time differences may be associated with an increase in the similarity coefficient after arrival optimization.

### 2.2. Arrival Refinement Based on Waveform Cross-Correlation

Many automatic processing workflows have already been developed for arrival-time picking using multi-trace cross-correlation, such as using over-determined linear equations to obtain the optimal result [27,30] or the iterative cross-correlation based method [10,33,34]. However, it is difficult to satisfy the conditions that both similarity coefficients of waveforms in the event and among events are greater than the threshold value in the actual data. The S/N and polarity of traces in the dataset are greatly different; therefore, a proper optimization strategy should be implemented to refine the arrival of low S/N events.

We propose an enhanced iterative cross-correlation method to calculate the residual time differences of multi-trace recordings. Initial arrival times generated by any of the picking algorithms are applied to align the microseismic recordings in this iterative cross-correlation method. To assist in the construction of the stacked trace, microseismic waveforms are then rescaled to equalize the pre-event noise level. Moreover, the time-shift (*τ*) is updated by correlating the stacked waveform with the recording after moveout correction. This process is repeated until the residual time shifts converge to less than a predefined threshold value (*ε*), indicating that the input data has been optimally realigned. The iterative cross-correlation workflow can be described in Figure 2.

The peak value of the cross-correlation function may be ambiguous due to the low S/N recording or the existence of multiple seismic phases. As a result, the position corresponding to the maximum value cannot dependably reflect the precise time delay. A windowed cross-correlation technique has been proposed to avoid the influence of the P-wave code [24]. In this study, we develop an enhanced cross-correlation approach by using a probability density function *f(t)* to restrict the time delay estimation as follows:(5)$$Ncc(t)=cc(t)\times f(t)=cc(t)\times \frac{1}{\sigma \sqrt{2\pi }}e^{-\frac{{\left(t-\mu \right)}^{2}}{2\sigma^{2}}}.$$
where *μ* is the mean value of the time difference (usually 1–2 times the dominant period of the signal), and *σ* is the estimated residual time. Figure 3 shows the comparison of the time difference estimation results between the normalized cross-correlation function and the proposed cross-correlation function. Figure 3a is one microseismic recording after moveout correction using the initial arrivals. Figure 3b,c are the recordings after residual time correction by the normalization cross-correlation function and the proposed cross-correlation function. The red circle in Figure 3b indicates the wrong time difference correction. It can avoid the large deviation in time difference estimation if the error is within a certain range (shown in Figure 3c,e).

### 2.3. Global Consistency Processing

An observed seismogram is a convolution of the source term, path effects, site effects, and instrumental responses. Event pairs with high cross-correlation coefficient values are considered to have close hypocenters and similar focal mechanisms. The waveform similarity characteristic of the events recorded by downhole arrays is primarily embodied by two aspects: (1) microseismic signals recorded by the adjacent receivers show waveforms similar on the records; (2) the waveforms of different microseismic events with nearby source locations and similar focal mechanisms show similar on the records. To improve the consistency of arrival time data and eliminate the influence of noise interference or abnormal situations of monitoring equipment, we propose a stepwise processing method for the global optimization of arrival picking for microseismic events.

The initial arrival picks *t_ik_* (*k* is the event number and *i* is the receiver number) of the microseismic data are obtained using an automatic algorithm, such as the STA/LTA or AIC methods. The workflow of the proposed method begins by optimizing intra-event picks. We apply the proposed enhanced iterative cross-correlation method to refine intra-event arrivals and superpose the aligned waveforms of each event after residual time correction. This process improves the relative picking accuracy between receivers and ensures the consistency of the intra-event arrival time data.

Global optimization is applied to calculate the inter-event residual time difference from the stacking waveforms of each event. Before optimization, the three-component stacking waveforms are rotated into one component to maximize the P-wave or S- wave signal energy. The arrival times after global optimization can be expressed
(6)$${T^{\prime }}_{ik}=t_{ik}+\Delta t_{ik}+\Delta T_{k}.$$
where *T*′ is the arrival time results after the global optimization. Δ*t_ik_* and Δ*T_k_* represent the intra-event and inter-event arrival corrections, respectively.

The workflow of the proposed global optimization method for microseismic event arrivals in this paper can be described as follows (Figure 4).

## 3. Synthetic Data Analysis

To validate the proposed method, we apply it to synthetic data generated using the geometry of a real microseismic monitoring system. The acquisition geometry consists of 15 receivers ranging in depth from 2443 to 2673 m. The spacing interval between two neighboring receivers is 10 m for levels 1 to 4 and levels 13 to 15, and 20 m for levels 4 to 13. The layout of this microseismic monitoring survey is shown in Figure 5. The velocity model used in this case is initially developed from the sonic logging data of the perforation well (as shown in Figure 5c). 100 microseismic events with random locations near one fracturing stage are used to generate synthetic one component recordings. The direct wave travel times are calculated using the layer velocity model. Ricker wavelets with dominant frequencies of 200 Hz and 100 Hz are used as the P and S-waves, respectively. The wavelet is multiplied by different weighting factors to simulate the various magnitudes of events, and these factors are exponentially decreased with traces to simulate the attenuation caused by the different propagation paths. Gaussian white noise is added to the 100 recordings with a sampling interval of 0.5 ms.

We use the short-term average/long-term average method (STA/LTA) to obtain the arrivals of each event, and then apply the improved iterative cross-correlation method to optimize the initial arrivals, achieving the adjusted picks after intra-event refine processing and inter-event refine processing sequentially. The number of samples in short and long-time windows are 20 and 100, respectively. To avoid incorrect arrival picks, we combine the limitations of the time difference between different receivers and different seismic phases when judging the P and S-wave arrivals. After intra-event residual time correction, the aligned waveforms of each event are superposed and then used for global optimization to obtain the inter-event residual time discrepancies. Figure 6 shows an example of microseismic event recording and the recordings after moveout correction using different arrivals. The residual time differences are accurately estimated, and the waveform consistency of the iterative cross-correlation refinement method is greater than that of the STA/LTA method, especially the P-wave recording in the example. Figure 7 shows the recordings of the stacked traces using the arrivals of different methods. The arrival differences among events are minimized after global optimization.

Figure 8 depicts the error comparison between the arrivals of three processing methods and the actual arrivals. In comparison to the arrival error of the STA/LTA method, the arrival error after intra-event optimization processing is significantly reduced, and the arrival error of the global optimization method decreases further. The bias in arrival picks (about 2~5 sampling numbers) may be caused by the shape of the synthetic microseismic signal, and it is in an acceptable range. In the actual data, the location of the arrival is unknown. The waveform similarity coefficient after moveout correction is a useful tool to analyze the quality of arrival results. As shown in Figure 9, the similarity coefficients of the moveout corrected waveforms are calculated using the true arrivals and arrivals of the STA/LTA method, after intra-event optimization processing. The similarity coefficients of moveout corrected waveforms after intra-event processing tend to approach those of true arrivals, indicating that arrival errors have been decreased.

## 4. Field Data Analysis

To demonstrate the performance of the proposed method, we used a field microseismic dataset during the 11-stage hydraulic fracture treatment which was shown in Figure 5. The dataset is from a tight reservoir fracturing monitoring project of Shengli Oilfield in eastern China. The total duration of the monitoring dataset is more than 27 h, and its sampling interval is 0.5 ms. Before the hydraulic fracturing stimulation starts, a perforation shot is fired in a third well located approximately 920 m northwest of the monitoring well. We used the signal of the perforation shot to orient the receivers to obtain the component-rotated data, and then we adopted a band-pass filter with cutoff frequencies 30 Hz and 300 Hz to eliminate low- and high-frequency noises from the continuous downhole monitoring recordings. In the continuous dataset, 521 triggered events were detected using an intra-event coherence-based event detection method, and their P- and S- wave arrival times were determined using the joint STA/LTA-polarization-AIC method [19].

Figure 10 shows four of these field microseismic events and their P- and S-wave arrival picks. The results seem to be accurate and acceptable. However, due to the difference between S/N and the waveform patterns, some of these picks are not accurate; they can be observed as local inconsistencies between traces both in common event gathers and common-receiver gathers after moveout correction based on pick times. Local waveform misalignment indicates the picking error among traces. Figure 11 and Figure 12 show common event gathers of the P-wave and S-wave of four microseismic events after moveout correction using the arrivals by the STA/LTA-polarization-AIC picker. The arrival picking errors are indicated by the local inconsistencies between waveforms (noted by the black arrows). In addition, the signal-to-noise ratios (SNR) of waveforms from each receiver are revealed to be considerably varied. The amplitude of the P-wave in Figure 11 decreases for deeper receivers as they may have been affected by radiation patterns and propagation paths [35]. Figure 13 displays four common-receiver gathers (CRG) after moveout correction, which are waveforms acquired by the same receiver (Receiver No.2, 5, 6, and 8) and can provide another perspective for finding erroneous picks.

The arrival times obtained from the joint STA/LTA-polarization–AIC method are utilized as initial picks for cross-correlation-based methods that refine the initial picks via the waveform similarity. Figure 14 and Figure 15 show the moveout corrected waveforms after the refinement of intra-event arrivals and the 3C stacked traces on their right. The adjustment of arrival time picks by multilevel algorithms results in more consistent waveforms than the waveforms shown in Figure 11 and Figure 12. Figure 16 shows the similarity coefficients of the moveout corrected waveforms before and after intra-event refinement. The similarity coefficients of the P wave (Figure 16a) and S wave (Figure 16b) are significantly increased for the majority of the events. The low similarity coefficients of the S wave in the z-component are due to the low energy of the S wave.

Although the waveform consistency after intra-event residual time difference correction has increased greatly, errors among events cannot be detected from the analysis of individual event recordings. The optimization process is not accomplished since there are still conspicuous time differences among events, which are observed as waveform misalignment in stacked waveform gathers, as shown in Figure 17. To limit the influence of the focal mechanism, these three-component recordings of the stacked waveforms of the P wave and S-waves were rotated to one component to maximize waveform energy and then processed by energy balancing and polarity unification. The superposed traces from events provide the arrival time and waveform shape of microseismic signals as well as enhancing the S/N, which is beneficial to calculate residual time discrepancies among events. Since arrival refinement is only applied in each event and does not take into consideration the interconnectivity of various events, the consistency of arrival picks among events varies even among waveforms with a high signal-to-noise ratio.

Figure 18 shows the stacked waveforms of the P wave and S wave after inter-event optimization processing. Compared with the waveforms of the single event-based processing, the waveforms after global optimization are more consistent, as shown in Figure 19. As shown in Figure 20, the similarity coefficients of the stacking waveforms are calculated using the adjusted arrivals of intra-event optimization processing, and the global optimization method. It demonstrates that after global optimization, the similarity coefficients of the stacked waveforms greatly increase, indicating that arrival discrepancies among the events are reduced.

## 5. Conclusions

We have proposed a quality assessment method based on waveform similarity coefficients for evaluating arrival results and a global optimization approach to refining the original arrival-time picks. The similarity coefficients of the moveout corrected waveforms and the stacking waveforms of each event have been calculated to evaluate the arrival results. The presence of arrival error in the conventional picking method diminishes the similarity coefficients, and the consistency among events is ignored. Iterative cross-correlation is utilized to estimate the residual time difference between traces, and a probability density distribution of the time difference is considered in the waveform cross-correlation to avoid wrong picks. The arrival discrepancies among the events are also obtained using the enhanced iterative cross-correlation approach on the stacked waveforms. The global optimization method has been tested on synthetic data and field data; the aligned waveforms indicate a greatly improved picking consistency of the arrival time result. The actual data processing results show that the arrival time data of microseismic events with a similar waveform from different fracturing stages can be regarded as a whole, which can rectify shortcomings in the consistency among the original results using traditional methods, thus improving the accuracy and consistency of the arrival times.

## Figures and Tables

**Figure 1 sensors-22-04065-f001:**
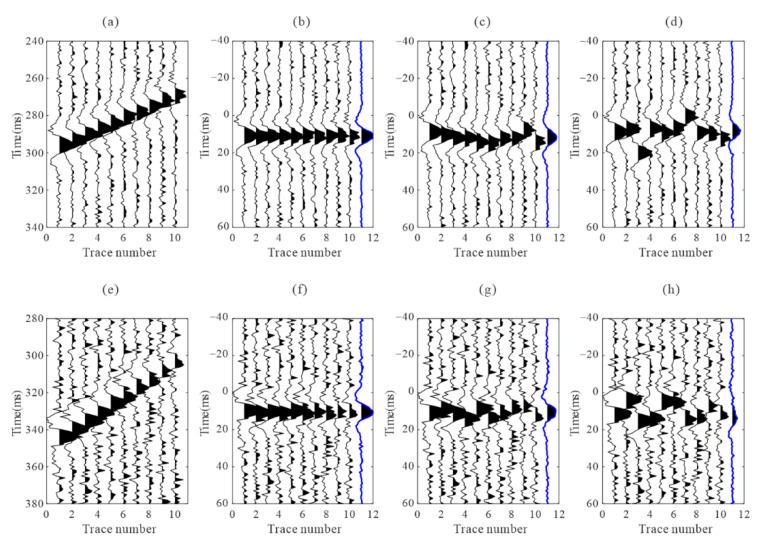
Two synthetic microseismic signals and the recordings after moveout correction using arrivals with different error levels. (**a**,**e**) are the synthetic recordings; (**b**,**f**) are the recordings after moveout correction using accurate arrivals; (**c**,**g**) are the recordings after moveout correction using arrivals with 2 ms standard deviation error; (**d**,**h**) are the recordings after moveout correction using arrivals with 5 ms standard deviation error. The blue traces are the stacked waveforms.

**Figure 2 sensors-22-04065-f002:**
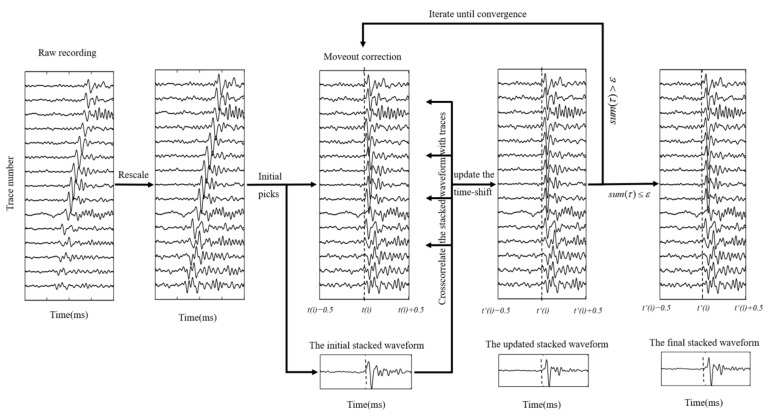
Iterative cross-correlation based workflow for refining arrival picking.

**Figure 3 sensors-22-04065-f003:**
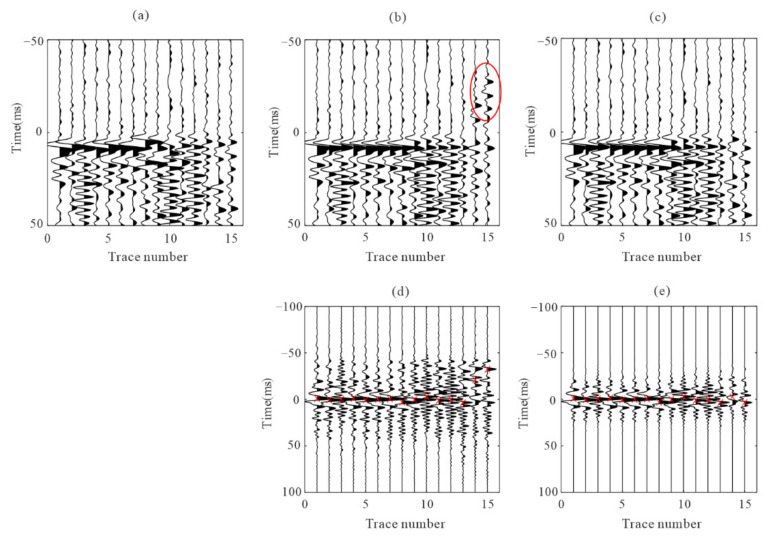
Waveforms after moveout correction based on adjusted picks from the time delay estimation. (**a**) the recording flattened using P pick times of one microseismic event example; (**b**,**c**) represent the recordings after adjusted picks correction using the normalized cross-correlation function and the proposed cross-correlation function, respectively; (**d**,**e**) are the normalized cross-correlation function and the proposed cross-correlation function, respectively. The red ‘x’ indicates the position of the maximum value. The red circle shows the wrong residual time estimation.

**Figure 4 sensors-22-04065-f004:**
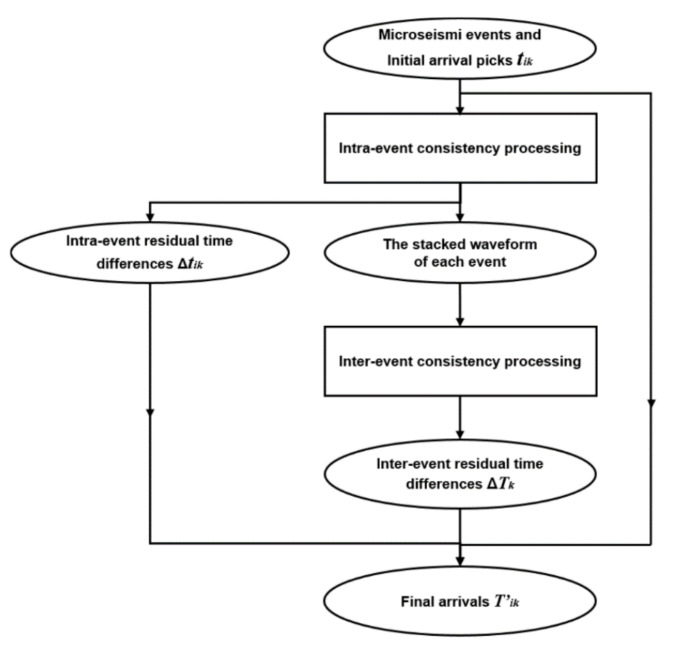
Flowchart of the proposed global optimization method.

**Figure 5 sensors-22-04065-f005:**
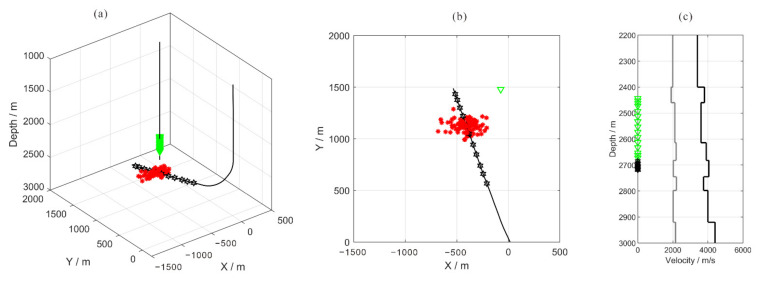
Layout and location of a real case microseismic monitoring system: the green triangles represent receivers, the black stars represent 11 fracturing sections, and the red stars represent the 100 microseismic event locations used to generate synthetic data. (**a**) 3-D view; (**b**) planar view; (**c**) the velocity model. Black and gray lines represent P and S wave velocities, respectively.

**Figure 6 sensors-22-04065-f006:**
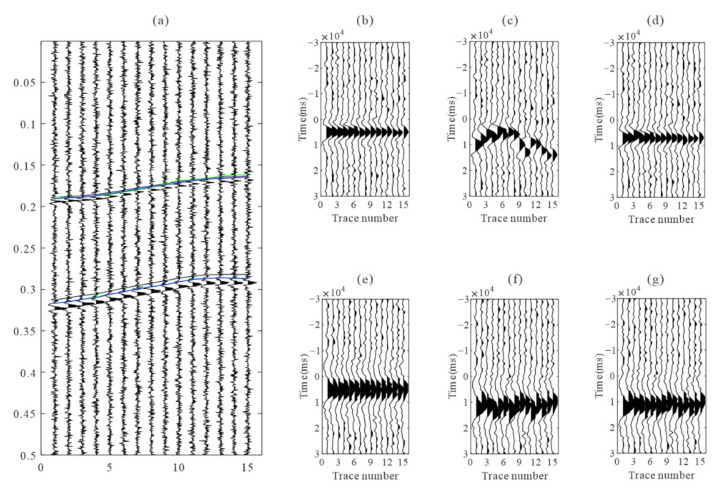
(**a**) synthetic microseismic recordings and the arrival picks. The black, green, and blue lines represent the true arrivals, the arrivals of STA/LTA, and intra-event optimization processing, respectively. (**b**,**e**) are P and S-waves after moveout correction using true arrivals; (**c**,**f**) are P and S-wave after moveout corrections using arrivals by STA/LTA; (**d**,**g**) are P and S-wave after moveout correction using refined picks by intra-event optimization processing.

**Figure 7 sensors-22-04065-f007:**
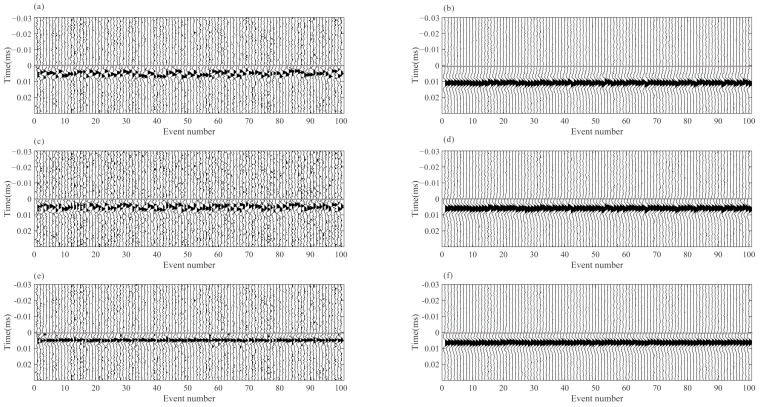
The stacked waveforms of each event obtained by different arrival picks. (**a**,**b**) are the P and S-wave stacked waveform using arrivals from STA/LTA; (**c**,**d**) are the P and S-wave stacked waveform using arrivals from intra-event optimization processing; (**e**,**f**) are the P and S-wave stacked waveform using arrivals from global optimization processing. The horizontal gray line represents the location of arrivals, and the recordings are processed by energy balancing between traces for better visibility.

**Figure 8 sensors-22-04065-f008:**
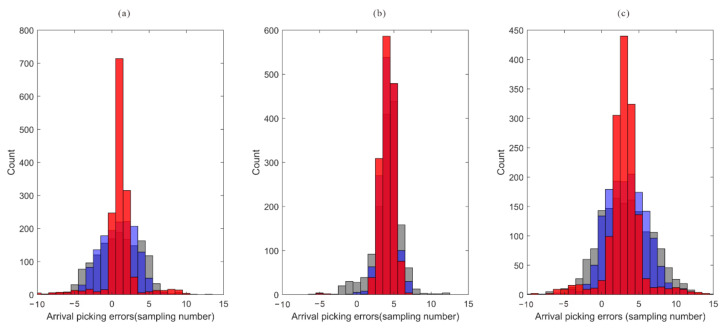
Arrival error histogram of different picking methods. (**a**) P wave arrival errors; (**b**) S wave arrival errors; (**c**) P-S time difference errors. The gray, blue, and red bars represent true arrivals and the arrivals of the STA/LTA method, the intra-event optimization processing method, and the inter-event optimization processing methods, respectively.

**Figure 9 sensors-22-04065-f009:**
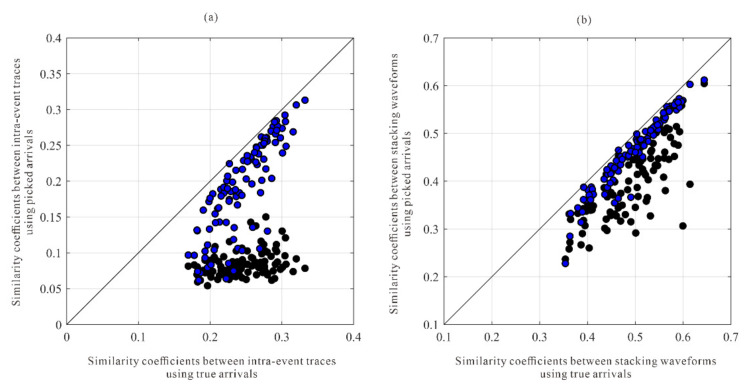
Quality assessments for the arrival results of different picking methods compared with the true arrivals. (**a**) the P wave similarity coefficients of intra-event traces; (**b**) the S wave similarity coefficients of intra-event traces. The black and blue circles represent the similarity coefficients calculated by those arrivals of the STA/LTA method and intra-event processing methods, respectively.

**Figure 10 sensors-22-04065-f010:**
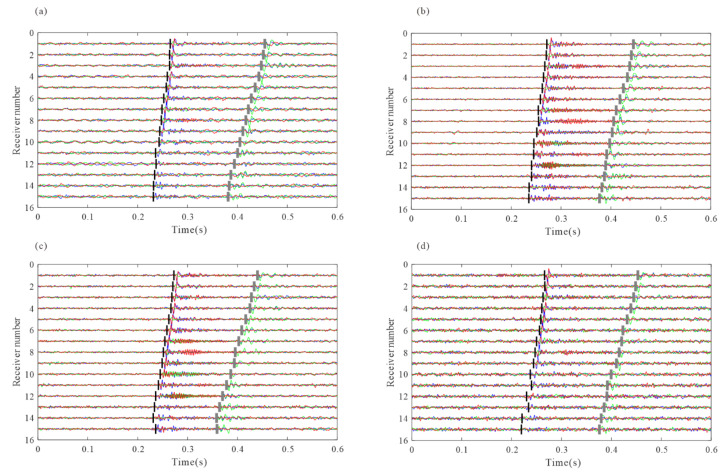
Four microseismic events and their P- and S-wave arrival picks. The balck vertical bars denote the P-wave arrival picks, and the gray ones denote those of the S-wave. Blue, green, and red lines represent rotated x-, y-, and z-components, respectively. (**a**) Event No.1; (**b**) Event No.2; (**c**) Event No.3; (**d**) Event No.4.

**Figure 11 sensors-22-04065-f011:**
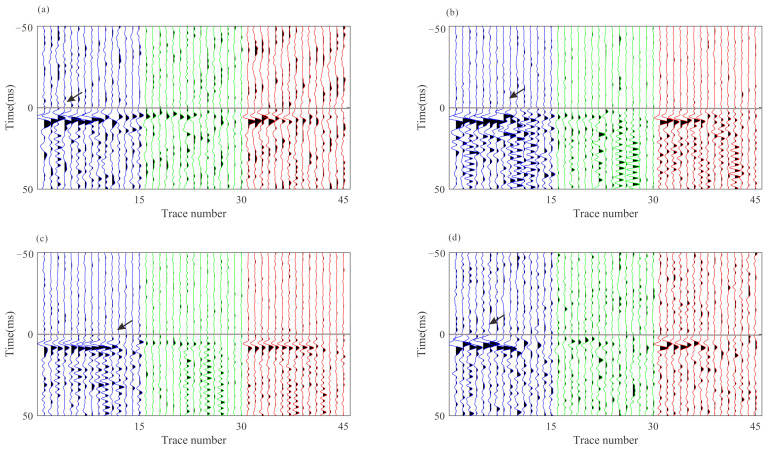
P-waves of four microseismic events recorded by 15 receivers after moveout correction. The horizontal gray line represents P-wave arrivals. (**a**–**d**) are P-wave recordings from the four events in Figure 10. Blue, green, and red lines represent rotated x-, y-, and z- components, respectively. The arrows indicate the local inconsistencies between waveforms.

**Figure 12 sensors-22-04065-f012:**
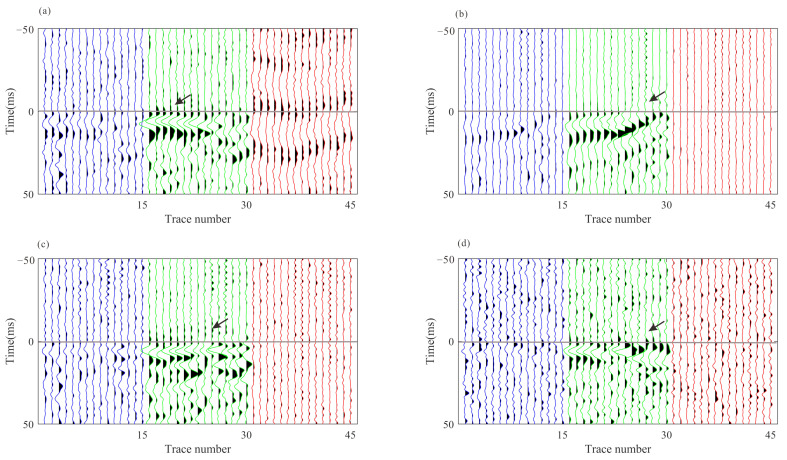
S-waves of four microseismic events recorded by 15 receivers after moveout correction. The horizontal gray line represents S-wave arrivals. (**a**–**d**) are S-wave recordings from the four events in Figure 10. Blue, green, and red lines represent rotated x-, y-, z- components, respectively. The arrows indicate the local inconsistencies between waveforms.

**Figure 13 sensors-22-04065-f013:**
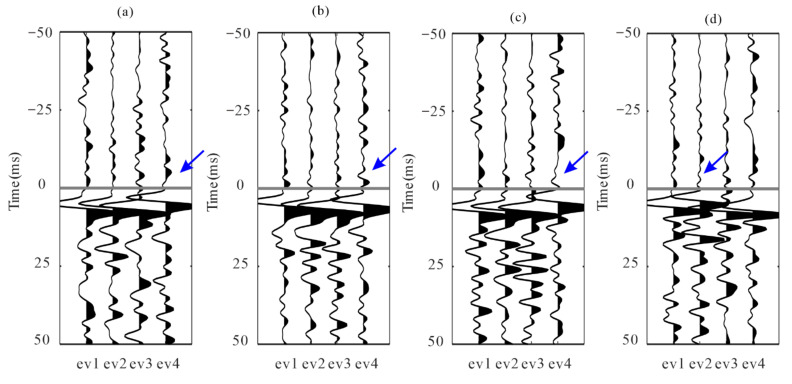
Common-receiver gathers of *x*-components flattened using P pick times. The horizontal gray line represents the location of P-wave arrivals. (**a**) Receiver No.2; (**b**) Receiver No.5; (**c**) Receiver No.6; (**d**) Receiver No.8. The blue arrows indicate the local inconsistencies between waveforms from different events.

**Figure 14 sensors-22-04065-f014:**
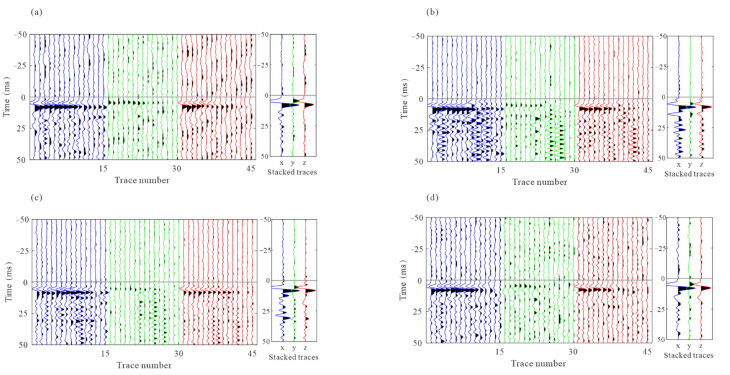
The 3C recording of the example events after residual time correction based on corrected P-wave arrival picks and the stacked waveforms on their right. (**a**–**d**) are P-wave recordings from the four events in Figure 10. The gray horizontal lines represent the corrected arrival picks of the P-wave. Blue, green, and red lines represent x-, y-, and z- components, respectively.

**Figure 15 sensors-22-04065-f015:**
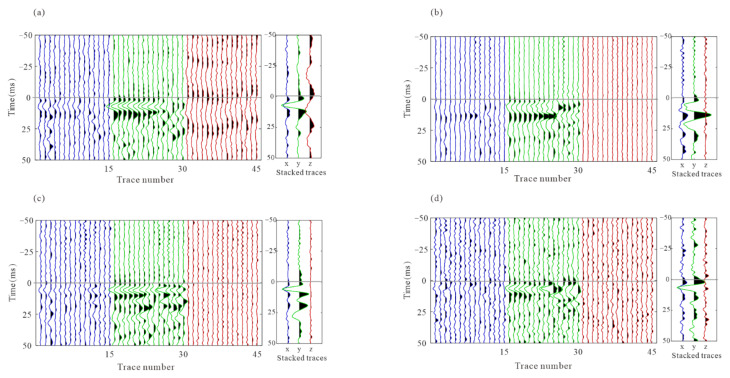
The 3C recording of the example events after residual time correction based on corrected S-wave arrival picks and the stacked waveforms on their right. (**a**–**d**) are S-wave recordings from the four events in Figure 10. The gray horizontal lines represent the corrected arrival picks of the S-wave. Blue, green, and red lines represent x-, y-, and z- components respectively.

**Figure 16 sensors-22-04065-f016:**
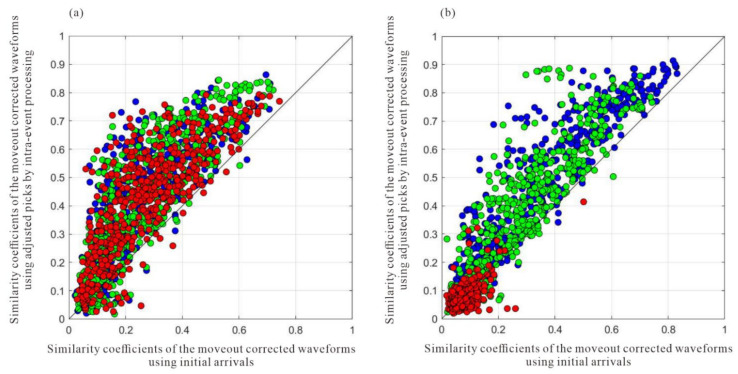
The similarity coefficients of moveout corrected waveforms before and after intra-event arrival refinement. (**a**) P wave; (**b**) S wave. Blue, green, and red circles represent x-, y-, and z-components respectively.

**Figure 17 sensors-22-04065-f017:**
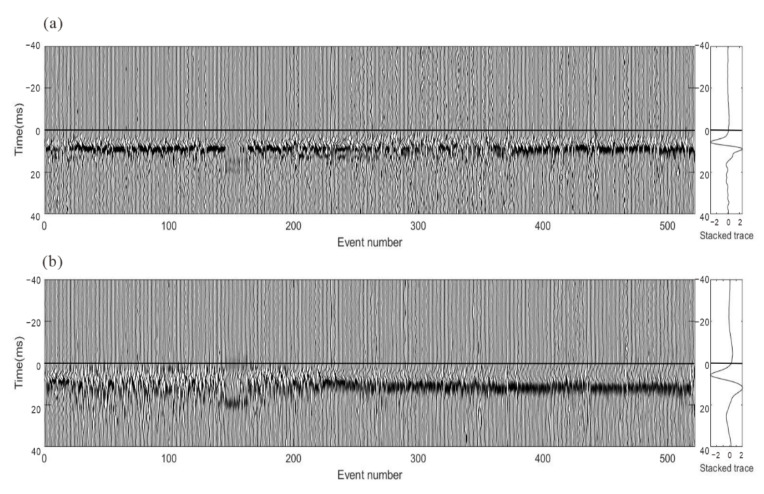
The stacked waveform gathers of all events based on adjusted picks. The horizontal lines show the arrival times of the stacked waveforms. (**a**) P wave; (**b**) S wave. The recordings are processed by energy balancing and polarity unification.

**Figure 18 sensors-22-04065-f018:**
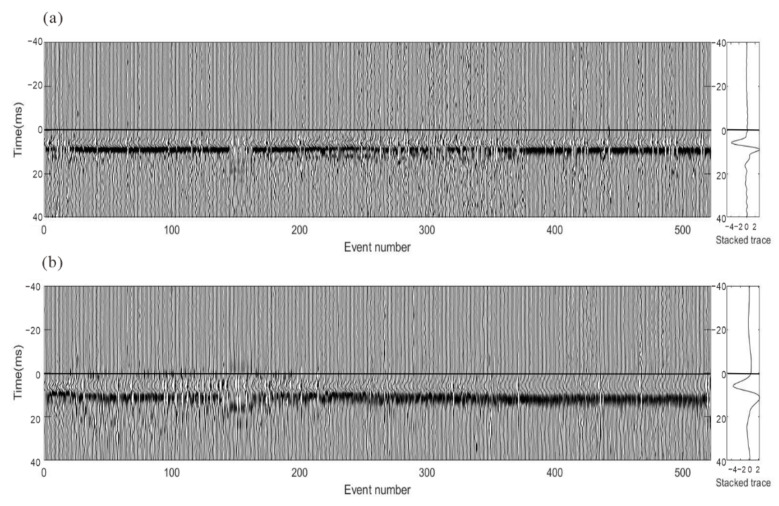
The stacked waveform gathers of all events after inter-event event refinement. The horizontal lines show the arrival times of the stacked waveforms. (**a**) P wave; (**b**) S wave. The recordings are processed by energy balancing and polarity unification.

**Figure 19 sensors-22-04065-f019:**
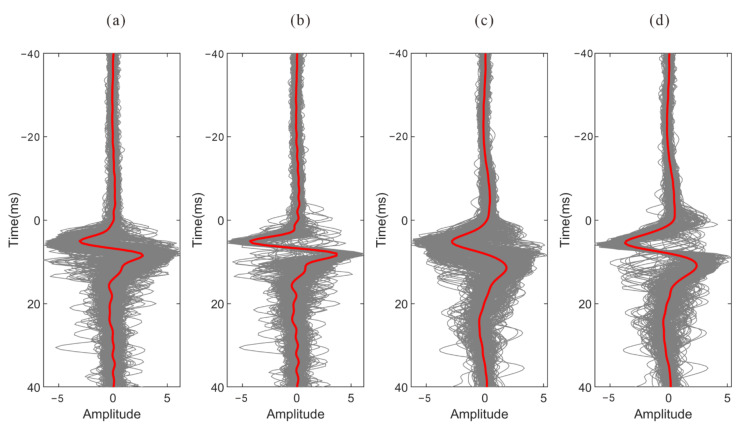
The stacked waveforms of all events before and after inter-event refinement. (**a**) P wave stacked waveforms before inter-event refinement; (**b**) P wave stacked waveforms before inter-event refinement; (**c**) S wave stacked waveforms before inter-event refinement; (**d**) S wave stacked waveforms after inter-event refinement. The red lines are the final stacked waveforms.

**Figure 20 sensors-22-04065-f020:**
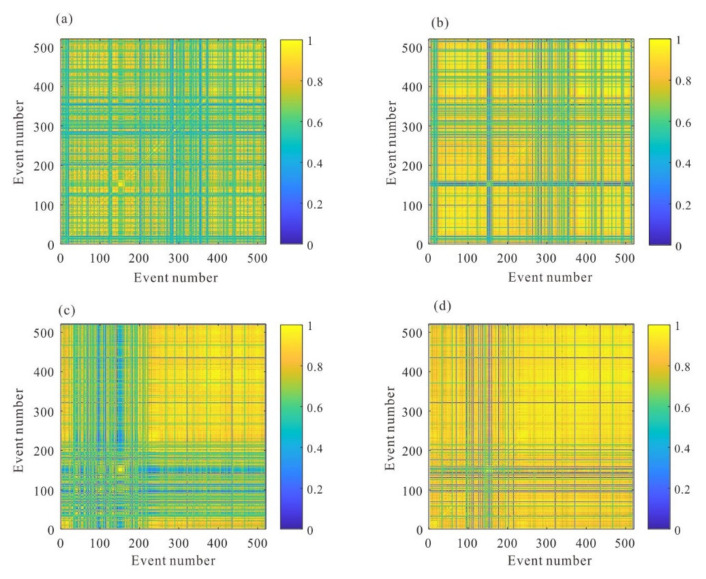
The similarity coefficients between the stacking waveforms before and after global optimization processing. (**a**) the similarity coefficients of the P wave before global optimization; (**b**) the similarity coefficients of the P wave after global optimization; (**c**) the similarity coefficients of the S wave before global optimization; (**d**) the similarity coefficients of the S wave after global optimization.

**Table 1 sensors-22-04065-t001:** Quality Assessment for the arrivals with various error levels.

The Similarity Coefficients	Accurate Arrivals	Arrival Error with 2 ms Standard Deviation Error	Arrivals Error with 5 ms Standard Deviation Error
*S_1_*	0.782	0.532	0.191
*S_2_*	0.576	0.471	0.163
*X_12_*	0.965	0.895	0.621

## Data Availability

Not applicable.

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
