# Peer review of "Optimization and Quality Assessment of Arrival Time Picking for Downhole Microseismic Events"

_sensors, 2022, doi:10.3390/s22114065_

Round 1

Reviewer 1 Report

Dear Authors

Your manuscript presents an interesting approach to the moveout problem of microseismic events. However, I have some concerns.

  • Firstly, there is no description of real field case data, where it has collected and what was the geology setting
  • Broader discussion with literature is required.
  • Language correction is required

You can find my comments and more questions in the attached pdf. 

Author Response

We greatly appreciate all your good suggestions for the manuscript. In the revised manuscript, we have tried our best to address all the comments and follow all the suggestions. We would like to take this opportunity to thank your valuable suggestions that help us to improve the quality of this manuscript. Below are our responses to your comments marked on our submission. More detailed revisions can be found in our revised manuscript.

Reviewer 2 Report

Minor comments

  1. A more comprehensive introduction should be provided, such as studies on cross-correlation based arrival time picking methods.
  2. The P-wave code will have an influence on the cross-correlation based arrival time picking methods, Wang et al (IEEE Access, 2020) proposed a windowed cross-correlation technique to avoid this.
  3. The P-wave arrival time system error (e.g., Shang et al, MSSP, 2022) should be corrected before an absolute or cross-correlation based arrival time picking.
  4. Lines 263~265 are not consistent with the Figure 9b, where the intra-event processing has a similarity coefficient to the STA/LTA method instead of the global optimization.

Author Response

We greatly appreciate all your good suggestions for the manuscript. Below are our responses to your comments marked on our submission. More detailed revisions can be found in our revised manuscript.

Reviewer 3 Report

Please check the word "assemble" in line № 106. It may be used in an inappropriate context.

Recommendations:

  • to extend the backgroud section.
  • to extend the introduction section and add a discussion of the application of the proposed method 

Author Response

(The authors gave the same response as above.)

Reviewer 4 Report

The paper deals with a very critical issue which is the accurate picking of arrival times of microseismic events. The problem is properly defined and a multilevel strategy is proposed to improve the picking accuracy and to assess the quality of the results. The strategy is validated on both synthetic and real data showing consistent results.

The paper is almost ready for publication. It only needs minor improvements, mostly regarding the discussion of the global optimization algorithm, as suggested in the following.

Line 32. I suggest to add the following citation:

  • Zhang, Z., Arosio, D., Hojat, A., Zanzi, L., 2021, Reclassification of Microseismic events through Hypocenter Location: Case Study on an Unstable Rock Face in Northern Italy, Geosciences, 11 (37), doi:10.3390/geosciences11010037.

Equation 4. Use wi(n) and wj(n) rather than wi and wj.

Line 156. I think that noise equalization might be appropriate but I suggest to discuss the benefit.  

Figure 2. It is very simple. It can be more informative if pre-event noise equalization and convergence test are included in the workflow.

Line 171 and following lines…change normalization with normalized. I think that the proper name is: normalized crosscorrelation function.

Line 176-177. Rephrase to improve the English and clarify the concept.

Lines 187-189. These lines seem to contain two separate statements. Substitute the comma in line 188 with a full stop.

Equation numbering: all the equations after equation 3 are numbered with the same number: 4.

Equation at line 208 (supposed to be equation 6) and Figure 4. I am not sure to understand the global optimization procedure and its impact on the final results. First question is about Figure 4. I see two large loops at the left side and right side of the central line of the flowchart. Are they real loops or only different exit points from which we can take the results to compare the quality of the picking produced with different levels of optimization (i.e., no optimization, intra-event optimization only, intra-event plus inter-event optimization)? Second question is about the importance of inter-event corrections. I understand from equation 6 that an inter-event correction produces an equal time shift applied to all the traces belonging to the same event. If this is true, what is the importance of this correction, i.e., what is the objective of this correction? To improve the accuracy in estimating the delay between P and S arrivals? Is there any other objective?

Line 241. Change Figure 8 with Figure 7.

Figure 7 and Figure 4. Zooming on waveforms in Figure 7, it seems to me that the effect of global optimization is not only a better alignment of the inter-events but also a modification of the stacked waveforms. Is it true? This makes me think that inter-event corrections are somehow influencing the stacking results after intra-event corrections. If this is true, I need to understand how this is happening. Are intra-event corrections revised after inter-event corrections? Flow chart in Figure 4 is not enough to describe the global optimization algorithm. Important details are missing in the methodology discussion. Are equations 3 and 4 introduced at page 3 used in the global optimization algorithm? How? Is the inter-event consistency processing in the flowchart equal to the intra-event consistency processing (i.e., based on an iterative application of crosscorrelation and stacking)? After reading the conclusions of the paper I would say yes but this should be made clear here.

Caption of Figure 7. What is the meaning of “the recordings are processed by energy balancing” in the caption? Is it referring to the trace normalization used to plot the data? Please clarify.

Lines 256-258. The bias motivation is not clear. Besides, it would be useful to mention what is the sampling interval in this simulation.

Figure 10. The quality of the picture is poor.

Lines 300-302. Rephrase to improve the English and clarify the concept.

Lines 300-320 and Figure 13. This discussion is not very clear. Try to improve showing examples from Figure 13.

Caption of Figure 15. Line 337. Change P-wave with S-wave.

Line 349. Clarify the meaning of energy balancing and polarity unification.

Author Response

We greatly appreciate all your good suggestions for the manuscript. In the revised manuscript, we have tried our best to address all the comments and follow all the suggestions. Below are our responses to your comments marked on our submission. More detailed revisions can be found in our revised manuscript.

Round 2

Reviewer 1 Report

Dear authors

Thank you for this interesting manuscript. I' am satisfied with article improvements. In my opinion it can be accepted in its current form. 

Best regards

Reviewer 2 Report

Accept.

Reviewer 4 Report

no more comments